# TAX-Pose: Task-Specific Cross-Pose Estimation for Robot Manipulation

**Chuer Pan,**[*] **Brian Okorn,**[*] **Harry Zhang,**[*] **Ben Eisner,**[*] **David Held**
Robotics Institute, School of Computer Science
Carnegie Mellon University, United States
`{chuerp, bokorn, haolunz, baeisner, dheld}@andrew.cmu.edu`

**Abstract:** How do we imbue robots with the ability to efficiently manipulate unseen objects and transfer relevant skills based on demonstrations? End-to-end learning methods often fail to generalize to novel objects or unseen configurations. Instead, we focus on the task-specific pose relationship between relevant parts of interacting objects. We conjecture that this relationship is a generalizable notion of a manipulation task that can transfer to new objects in the same category; examples include the relationship between the pose of a pan relative to an oven or the pose of a mug relative to a mug rack. We call this task-specific pose relationship "cross-pose" and provide a mathematical definition of this concept. We propose a vision-based system that learns to estimate the cross-pose between two objects for a given manipulation task using learned cross-object correspondences. The estimated cross-pose is then used to guide a downstream motion planner to manipulate the objects into the desired pose relationship (placing a pan into the oven or the mug onto the mug rack). We demonstrate our method's capability to generalize to unseen objects, in some cases after training on only 10 demonstrations in the real world. Results show that our system achieves state-of-the-art performance in both simulated and real-world experiments across a number of tasks. Supplementary information and videos can be found on our [project website](#).

**Keywords:** Learning from Demonstration, Manipulation, 3D Learning

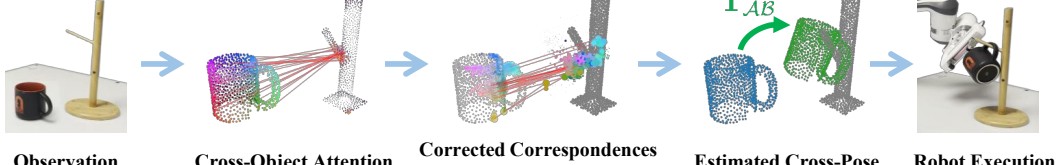

Figure 1: To solve a relative placement task, TAX-Pose uses cross-object attention to estimate dense cross-object correspondences and importance weights for each object point. This dense estimate is mapped to a single "cross-pose" which the robot uses to accomplish the given task.

## 1 Introduction

Many manipulation tasks require a robot to move an object to a location relative to another object. For example, a cooking robot may need to place a lasagna in an oven, place a pot on a stove, place a plate in a microwave, place a mug onto a mug rack, or place a cup onto a shelf. Understanding and placing objects in task-specific locations is a key skill for robots operating in human environments. Further, this skill should generalize to novel objects within the training categories, such as placing new trays into the oven or new mugs onto a mug rack. A common approach in robot learning is to train a policy "end-to-end," mapping from pixel observations to low-level robot actions. However, end-to-end trained policies cannot easily reason about complex pose relationships such as the ones described above, and they have difficulty generalizing to unseen object configurations.

---

[*]Equal Contribution. See Appendix H for a detailed list of each author's contributions.

6th Conference on Robot Learning (CoRL 2022), Auckland, New Zealand.

In contrast, we propose a method that learns to reason about the 3D geometric relationship between a pair of objects. For the type of tasks defined above, the robot needs to reason about the relationship between key parts on one object with respect to key parts on another object. For example, to place a mug on a mug rack, the robot must reason about the relationship between the pose of the mug handle and the pose of the mug rack; if the mug rack changes its pose, then the pose of the mug must change accordingly in order to still be placed on the rack (see Figure 3). We name this task-specific notion of the pose relationship between a pair of objects as "cross-pose" and we formally define it mathematically. Further, we propose a vision system that can efficiently estimate the cross-pose from a small number of demonstrations of a given task, generalizing to novel objects within the training categories. To complete the manipulation task, we use the estimated cross-pose as the target of a motion planning algorithm, which will move the objects into the desired configuration (e.g. placing the mug onto the rack, placing the lasagna into the oven, etc).

In this paper, we present TAX-Pose (TAsk-specific Cross-Pose), a deep 3D vision-based method that learns to predict a task-specific pose relationship between a pair of objects from a set of demonstrations. Our cross-pose estimation system is provably translation equivariant and can generalize from a small number of real-world demonstrations (in some cases as few as 10) to new objects in unseen poses.

The contributions of this paper include:

1. A precise definition of "cross-pose," which defines a task-specific pose relationship between two objects.
2. A novel method that estimates soft-correspondences between two objects, from which the cross-pose between the objects can be estimated (see Figure 1); this method is provably translation equivariant and can learn from a small number of real-world demonstrations.
3. A robot system to manipulate objects into the desired cross-pose to achieve a given manipulation task.

We present simulated and real-world experiments to test the performance of our system in achieving a variety of relative placement manipulation tasks. We demonstrate our method on a semantic placement task, in which the robot must place an object in, on, or around a novel object (Figure 2, top). We also demonstrate our method on precise placement tasks, such as hanging a mug on a rack (Figure 2, bottom) or placing a bottle or bowl on a shelf; in both cases our method generalizes to new object configurations and new objects within the training categories.

**Observations**   **Robot Execution**

Figure 2: We study relative placement tasks, in which one object needs to be placed in a position relative to another object. Here are two of the tasks that we demonstrate our method on: **Top:** *PartNet-Mobility Placement Task* requires one object (e.g. a block) to be placed relative to another object (e.g. a drawer) by a semantic goal position (e.g. inside); **Bottom:** *Mug Hanging Task* requires placing the mug's handle on the mug rack.

## 2   Related Work

**Object Pose Estimation**: Pose estimation is the task of detecting and inferring the 6DoF pose of an object, which includes its position and orientation, with respect to some previously defined object reference frame [1, 2, 3, 4, 5, 6]. Recent work [7, 8, 9, 10] proposed to use 3D semantic keypoints as an alternative form of object representation. While keypoint-based methods can generalize within an object class, they require a significant amount of hand annotated data or access to a simulated version of the task to learn to estimate the keypoint locations. In contrast, our method is able to learn from just 10 real-world demonstrations. Another approach is to use dense embeddings, such as Dense Object Nets (DON) [11] and Neural Descriptor Fields (NDF) [12], which achieve generalization across classes by predicting dense embeddings in the observation and matching them to embeddings of the demonstration objects. However, DON [11] and NDF [12] assume that the target object is moved relative to a static reference object in a "known canonical configuration" (e.g. the pose of the mug rack in NDF [12] is assumed to be known and fixed). In contrast, our method reasons about the geometric relationship between a pair of objects and hence does not need to assume

a static environment. Thus, for example, our method is able to perform the mug hanging task while varying the pose of the mug rack (see our project website), whereas the baselines (DON [11], NDF [12]) cannot. Further, we show that our method significantly outperforms both the DON [11] and NDF [12] baselines, especially when given a very small number of demonstrations.

**Point Cloud Registration**: Our method for estimating the cross-pose between two objects builds upon previous work in point cloud registration. The typical objective in point cloud registration is to find the optimal rigid alignment between two point clouds, to minimize the sum of squared distances between two sets of points. Traditionally, Iterative Closest Point (ICP) [13] and its variants [14, 15, 16, 17, 18, 19] have been used to compute the optimal rigid alignment between two point clouds. Deep Closest Point (DCP) [20] avoids local minima common for ICP by seeking to approximate correspondence in a high-dimensional learned feature space. Our method builds upon the architecture of DCP for cross-pose estimation; however, in contrast to point cloud registration, in which the objective is to minimize the sum of squared distances between two sets of points on the same object in two different poses, our objective is to estimate a task-specific pose relationship between two different objects. Extending the framework from DCP, we learn a residual to the soft correspondences, allowing for points to match outside the convex hull of each object. This component is necessary when computing soft correspondences between objects of drastically different morphologies (such as a mug and a mug rack).

## 3   Problem Statement

**Relative placement tasks:** In this paper, we are specifically interested in "relative placement tasks." Given two objects, $\mathcal{A}$ and $\mathcal{B}$, a "relative placement task" is the task of placing object $\mathcal{A}$ at a pose relative to object $\mathcal{B}$. For example, consider the task of placing a lasagna in an oven, placing a mug on a rack, or placing a robot gripper on the rim of a mug. All of these tasks involve placing one object (which we call the "action" object $\mathcal{A}$) at a semantically meaningful location relative to another object (which we call the "anchor" object $\mathcal{B}$)[2].

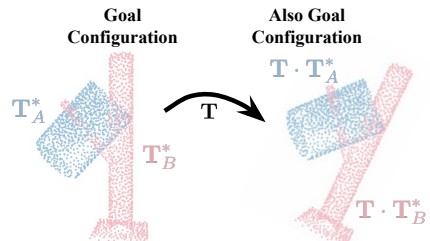

Figure 3: If we transform both the action object (mug) and the anchor object (rack) by the same transform, then the relative pose between these objects is unchanged (the mug is still "on" the rack) so the mug is still in the goal configuration.

Specifically, suppose that $\mathbf{T}^*_{\mathcal{A}}$ and $\mathbf{T}^*_{\mathcal{B}}$ are $SE(3)$ poses for objects $\mathcal{A}$ and $\mathcal{B}$ respectively (in a shared world reference frame[3]) for which a desired task is considered complete (lasagna is in the oven; mug is on the rack, etc). Then for a relative placement task, if objects $\mathcal{A}$ and $\mathcal{B}$ are in poses $\mathbf{T} \cdot \mathbf{T}^*_{\mathcal{A}}$ and $\mathbf{T} \cdot \mathbf{T}^*_{\mathcal{B}}$ (respectively) for any transform $\mathbf{T}$, then the task will also be considered to be complete, as seen in Figure 3. In other words, if $\mathbf{T}^*_{\mathcal{B}}$ represents the pose of the rack and $\mathbf{T}^*_{\mathcal{A}}$ represents the pose of the mug on the rack (at task completion); then if we transform the both the mug and rack poses by $\mathbf{T}$, then the mug will still be located on the rack. Formally, this property can be defined with the following Boolean function,

$$\text{RelPlace}(\mathbf{T}_{\mathcal{A}}, \mathbf{T}_{\mathcal{B}}) = \textbf{SUCCESS} \text{ iff } \exists \mathbf{T} \in SE(3) \text{ s.t. } \mathbf{T}_{\mathcal{A}} = \mathbf{T} \cdot \mathbf{T}^*_{\mathcal{A}} \text{ and } \mathbf{T}_{\mathcal{B}} = \mathbf{T} \cdot \mathbf{T}^*_{\mathcal{B}}. \quad (1)$$

For many real semantic placement tasks, there are actually sets of valid solutions which solve each task (i.e., there are many potential locations to place an object on a table to achieve a semantic "object-on-table" relationship). However, for this work, we consider precise placement tasks under the simplifying assumption that, for a given pose of object $\mathcal{B}$, there is a single, unambiguous pose of object $\mathcal{A}$ needed to achieve the task.

**Definition of Cross-Pose:** Given the above definition of a relative placement task, our goal will be to determine how to move object $\mathcal{A}$ so that it will be in the "goal pose," which, as described above, is defined relative to the pose of object $\mathcal{B}$. To achieve this, one option is to estimate the poses of objects $\mathcal{A}$ and $\mathcal{B}$ separately and then compute the transformation needed to move object $\mathcal{A}$ into the goal pose. However, the pose estimate of each object will have errors, and these errors will accumulate when the poses are combined into the single relative pose needed to reach the goal configuration.

---

[2]Note that the definition of action and anchor is symmetric; either object can be treated as the action object and the other as the anchor.

[3]All $SE(3)$ transformations in this work are defined in a fixed, arbitrary world frame.

Instead of estimating the pose of each object independently, we aim to learn a function $f(\mathbf{P}_\mathcal{A}, \mathbf{P}_\mathcal{B})$, which takes as input the point clouds $\mathbf{P}_\mathcal{A}$ and $\mathbf{P}_\mathcal{B}$ for both objects $\mathcal{A}$ and $\mathcal{B}$, where $\mathbf{P}_\mathcal{A} \in \mathbb{R}^{3 \times N_\mathcal{A}}$ and $\mathbf{P}_\mathcal{B} \in \mathbb{R}^{3 \times N_\mathcal{B}}$ are 3D point clouds of sizes $N_\mathcal{A}$ and $N_\mathcal{B}$, respectively. This function outputs an SE(3) rigid transformation, $f(\mathbf{P}_\mathcal{A}, \mathbf{P}_\mathcal{B}) = \mathbf{T}_{\mathcal{AB}}$, where we refer to $\mathbf{T}_{\mathcal{AB}}$ as the "cross-pose" between object $\mathcal{A}$ and object $\mathcal{B}$. For notational convenience, we occasional write $f$ as a function of the poses $\mathbf{T}_\mathcal{A}$, $\mathbf{T}_\mathcal{B}$ of point clouds $\mathbf{P}_\mathcal{A}$ and $\mathbf{P}_\mathcal{B}$ respectively (with respect to a global reference frame) such that $f(\mathbf{T}_\mathcal{A}, \mathbf{T}_\mathcal{B}) := f(\mathbf{P}_\mathcal{A}, \mathbf{P}_\mathcal{B})$. This notational change is to make the transformation math more intuitive; in practice, this function only ever receives point clouds as input.

We will define the cross-pose $\mathbf{T}_{\mathcal{AB}}$ (below) such that, if we transform object $\mathcal{A}$ by $\mathbf{T}_{\mathcal{AB}}$, then object $\mathcal{A}$ will be in the goal pose relative to object $\mathcal{B}$ for the relative placement task. For example, suppose that $\mathbf{T}_\mathcal{A}^*$ and $\mathbf{T}_\mathcal{B}^*$ are poses for objects $\mathcal{A}$ and $\mathcal{B}$, respectively, for which a desired relative placement task is considered complete. In this configuration, the cross-pose of these objects would be $f(\mathbf{T}_\mathcal{A}^*, \mathbf{T}_\mathcal{B}^*) = \mathbf{I}$ where $\mathbf{I}$ is the identity, as object $\mathcal{A}$ does not need to be moved to complete the task. Further, based on the definition of a relative placement task given above, if both objects are transformed by the same transform $\mathbf{T}$, then the objects will still be in the desired relative pose,

$$f(\mathbf{T} \cdot \mathbf{T}_\mathcal{A}^*, \mathbf{T} \cdot \mathbf{T}_\mathcal{B}^*) = f(\mathbf{T}_\mathcal{A}^*, \mathbf{T}_\mathcal{B}^*) = \mathbf{I} \tag{2}$$

for any transform $\mathbf{T} \in SE(3)$. Now, let us assume that objects $\mathcal{A}$ and $\mathcal{B}$ are not in the goal configuration and have pose $\mathbf{T}_\mathcal{A} = \mathbf{T}_\alpha \cdot \mathbf{T}_\mathcal{A}^*$ and $\mathbf{T}_\mathcal{B} = \mathbf{T}_\beta \cdot \mathbf{T}_\mathcal{B}^*$, respectively, for arbitrary transforms $\mathbf{T}_\alpha$ and $\mathbf{T}_\beta \in SE(3)$. We then define the "cross-pose" of objects $\mathcal{A}$ and $\mathcal{B}$ as:

$$f(\mathbf{T}_\mathcal{A}, \mathbf{T}_\mathcal{B}) = f(\mathbf{T}_\alpha \cdot \mathbf{T}_\mathcal{A}^*, \mathbf{T}_\beta \cdot \mathbf{T}_\mathcal{B}^*) = \mathbf{T}_{\mathcal{AB}} := \mathbf{T}_\beta \cdot \mathbf{T}_\alpha^{-1}. \tag{3}$$

Note that this definition is equivalent to Equation 2 for the special case of $\mathbf{T}_\alpha = \mathbf{T}_\beta$. This definition of cross-pose allows us to move object $\mathcal{A}$ into the goal configuration, relative to object $\mathcal{B}$:

$$\mathbf{T}_{\mathcal{AB}} \cdot \mathbf{T}_\mathcal{A} = (\mathbf{T}_\beta \cdot \mathbf{T}_\alpha^{-1}) \cdot (\mathbf{T}_\alpha \cdot \mathbf{T}_\mathcal{A}^*) = \mathbf{T}_\beta \cdot \mathbf{T}_\mathcal{A}^*, \tag{4}$$

satisfying the relative placement condition defined in Equation 1 with $\mathbf{T} = \mathbf{T}_\beta$. Alternatively, we could have instead transformed object $\mathcal{B}$ by the inverse of the cross-pose to achieve the task.

# 4 Method

**Overview:** We frame the task of cross-pose estimation as a soft correspondence-prediction task between a pair of point clouds, followed by an analytical least-squares optimization to find the optimal *cross-pose* for the predicted correspondences. As described in Appendix B, this correspondence-based approach allows our method to be translation-equivariant: translating either object ($\mathcal{A}$ or $\mathcal{B}$) will lead to a translated cross-pose prediction. This allows our method to automatically adapt to novel positions of both the anchor and action objects, unlike previous work which assumes a static anchor [12]. Our method for task-specific cross-pose estimation, known as TAX-Pose, consists of the following steps, as shown in Figure 4:

1. **Soft Correspondence Prediction**: For a pair of objects $\mathcal{A}, \mathcal{B}$, a neural network learns to predict a per-point embedding to establish a (soft) correspondence between $\mathcal{A}$ and $\mathcal{B}$, which are called

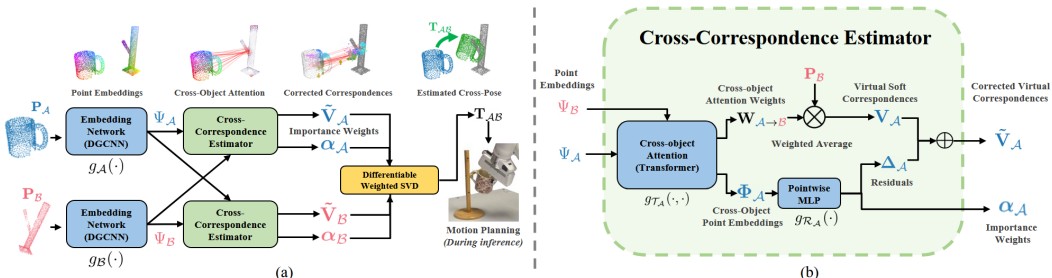

Figure 4: TAX-Pose Training Overview: Given a specific task, our method takes as input two point clouds and outputs the cross-pose between them needed to achieve the task. TAX-Pose first learns point clouds features using two DGCNN [21] networks and two Transformers [22]. Then the learned features are each input to a point residual network to predict per-point soft correspondences and weights across the two objects. The desired cross-pose can be inferred analytically from these correspondences using singular value decomposition.

"virtual soft correspondences." The corresponding points are constrained to be within the convex hulls of $\mathcal{B}$ and $\mathcal{A}$ respectively.

2. **Adjustment via Correspondence Residuals**: For most estimation tasks, some points in object $\mathcal{A}$ may not be within the convex hull of object $\mathcal{B}$; for instance, when a mug is placed on a mug rack, most points on the mug will be outside of the convex hull of the mug rack. To accommodate these cases, we apply a pointwise residual vector to displace each of the predicted soft correspondences. These "corrected virtual correspondences" allow points in $\mathcal{A}$ to correspond to locations in free space near $\mathcal{B}$.

3. **Find the Optimal Transform**: Because the cross-pose is defined as a rigid transformation of object $\mathcal{A}$, we use a differentiable weighted SVD to find the transformation that minimizes the weighted least squares difference to the corrected virtual correspondences.

Because each step above is differentiable, the whole model can be optimized end-to-end, despite having an interpretable internal structure which we describe below. Our method is heavily inspired by Deep Closest Point (DCP) [20]. The key difference between our pose alignment model and DCP is that we predict the cross-pose between two *different* objects for a given task instead of registering two point clouds of an identical object. Additionally, TAX-Pose can predict relationships where these clouds may not have any points of contact or overlap.

We now describe our cross-pose estimation algorithm in detail. To recap the problem statement, given objects $\mathcal{A}$ and $\mathcal{B}$ with point cloud observations $\mathbf{P}_\mathcal{A} \in \mathbb{R}^{3 \times N_\mathcal{A}}$, $\mathbf{P}_\mathcal{B} \in \mathbb{R}^{3 \times N_\mathcal{B}}$ respectively, our objective is to estimate the task-specific cross-pose $\mathbf{T}_{\mathcal{AB}} = f(\mathbf{P}_\mathcal{A}, \mathbf{P}_\mathcal{B}) \in SE(3)$. Note that the cross-pose between object $\mathcal{A}$ and $\mathcal{B}$ is defined with respect to a given task (e.g. putting a lasagna in the oven, putting a mug on the rack, etc).

### 4.1 Cross-Pose Estimation via Soft Correspondence Prediction

**Soft Correspondence Prediction**: The first step of the method is to compute two sets of correspondences between $\mathcal{A}$ and $\mathcal{B}$, one which maps from points in $\mathcal{A}$ to $\mathcal{B}$, and one which maps from points in $\mathcal{B}$ to $\mathcal{A}$. These need not be a bijection, and can be asymmetric. As we want each step to be differentiable, we follow DCP's conventions and estimate a *soft correspondence*. This assigns a *virtual soft corresponding point* $\mathbf{v}_i^\mathcal{A} \in \mathbf{V}_\mathcal{A}$ to every point $\mathbf{p}_i^\mathcal{A} \in \mathbf{P}_\mathcal{A}$ by computing a convex combination of points in $\mathbf{P}_\mathcal{B}$, and vice versa. Formally:

$$\mathbf{v}_i^\mathcal{A} = \mathbf{P}_\mathcal{B} \mathbf{w}_i^{\mathcal{A} \to \mathcal{B}} \quad \text{s.t.} \quad \sum_{j=1}^{N_\mathcal{B}} w_{ij}^{\mathcal{A} \to \mathcal{B}} = 1 \quad (5a) \quad \mathbf{v}_i^\mathcal{B} = \mathbf{P}_\mathcal{A} \mathbf{w}_i^{\mathcal{B} \to \mathcal{A}} \quad \text{s.t.} \quad \sum_{j=1}^{N_\mathcal{A}} w_{ij}^{\mathcal{B} \to \mathcal{A}} = 1 \quad (5b)$$

with normalized weight vectors $\mathbf{w}_i^{\mathcal{A} \to \mathcal{B}} \in \mathbf{W}_{\mathcal{A} \to \mathcal{B}}$ and $\mathbf{w}_i^{\mathcal{B} \to \mathcal{A}} \in \mathbf{W}_{\mathcal{B} \to \mathcal{A}}$. Importantly, these virtual corresponding points are not constrained to the surfaces of $\mathcal{A}$ or $\mathcal{B}$; instead, they are constrained to the convex hulls of $\mathbf{P}_\mathcal{B}$ and $\mathbf{P}_\mathcal{A}$, respectively.

To compute the weights $\mathbf{w}_i^{\mathcal{A} \to \mathcal{B}}$, $\mathbf{w}_i^{\mathcal{B} \to \mathcal{A}}$ in Equations 5a and 5b, we first encode each point cloud $\mathbf{P}_\mathcal{A}$ and $\mathbf{P}_\mathcal{B}$ into a latent space using a neural network encoder, DGCNN [21]. This encoder head is comprised of two distinct encoders $g_\mathcal{A}$ and $g_\mathcal{B}$, each of which receives point cloud $\mathbf{P}_\mathcal{A}$ and $\mathbf{P}_\mathcal{B}$, respectively, zero-centers them, and outputs a dense, point-wise embedding for each object (see Figure 4): $\mathbf{\Psi}_\mathcal{A} = g_\mathcal{A}(\bar{\mathbf{P}}_\mathcal{A}) \in \mathbb{R}^{N_\mathcal{A} \times d}$, $\mathbf{\Psi}_\mathcal{B} = g_\mathcal{B}(\bar{\mathbf{P}}_\mathcal{B}) \in \mathbb{R}^{N_\mathcal{B} \times d}$ where $\psi_i^\mathcal{K} \in \mathbf{\Psi}_\mathcal{K}$ is the $d$-dimensional embedding of the $i$-th point in object $\mathcal{K}$, and $\bar{\mathbf{P}}_\mathcal{K}$ is the zero-centered point cloud for object $\mathcal{K}$. Because we want the cross-correspondence to incorporate information about both point clouds, we then employ a cross-object attention module between the two dense feature sets to obtain *cross-object point embeddings*, $\mathbf{\Phi}_\mathcal{A} \in \mathbb{R}^{N_\mathcal{A} \times d}$ and $\mathbf{\Phi}_\mathcal{B} \in \mathbb{R}^{N_\mathcal{B} \times d}$, defined as:

$$\mathbf{\Phi}_\mathcal{A} = \mathbf{\Psi}_\mathcal{A} + g_{\mathcal{T}_\mathcal{A}}(\mathbf{\Psi}_\mathcal{A}, \mathbf{\Psi}_\mathcal{B}), \quad \mathbf{\Phi}_\mathcal{B} = \mathbf{\Psi}_\mathcal{B} + g_{\mathcal{T}_\mathcal{B}}(\mathbf{\Psi}_\mathcal{B}, \mathbf{\Psi}_\mathcal{A}) \quad (6)$$

where $g_{\mathcal{T}_\mathcal{A}}$, $g_{\mathcal{T}_\mathcal{B}}$ are Transformers [22].

Finally, recall that our goal was to compute a set of normalized weight vectors $\mathbf{W}_{\mathcal{A} \to \mathcal{B}}$, $\mathbf{W}_{\mathcal{B} \to \mathcal{A}}$. To compute the virtual corresponding point $\mathbf{v}_i^\mathcal{A}$ assigned to any point $\mathbf{p}_i^\mathcal{A} \in \mathbf{P}^\mathcal{A}$, we can extract the desired normalized weight vector $\mathbf{w}_i^{\mathcal{A} \to \mathcal{B}}$ from intermediate attention features of the cross-object attention module as:

$$\mathbf{w}_i^{\mathcal{A} \to \mathcal{B}} = \text{softmax} \left( \frac{\mathbf{K}_\mathcal{B} \mathbf{q}_i^\mathcal{A}}{\sqrt{d}} \right), \quad \mathbf{w}_i^{\mathcal{B} \to \mathcal{A}} = \text{softmax} \left( \frac{\mathbf{K}_\mathcal{A} \mathbf{q}_i^\mathcal{B}}{\sqrt{d}} \right) \quad (7)$$

where $\mathbf{q}_i^{\mathcal{K}} \in \mathbf{Q}_{\mathcal{K}}$, and $\mathbf{Q}_{\mathcal{K}}, \mathbf{K}_{\mathcal{K}} \in \mathbb{R}^{\mathbf{N}_{\mathcal{K}} \times d}$ are the query and key values (respectively) for object $\mathcal{K}$ associated with cross-object attention Transformer module $g_{\mathcal{T}_{\mathcal{K}}}$ (see Appendix C for details). These weights are then used to compute the virtual soft correspondences $\mathbf{V}_{\mathcal{A}}, \mathbf{V}_{\mathcal{B}}$ using Equation 5.

**Adjustment via Correspondence Residuals:** As previously stated, the virtual soft correspondences $\mathbf{V}_{\mathcal{A}}, \mathbf{V}_{\mathcal{B}}$ given by Equations 5a and 5b are constrained to be within the convex hull of each object. However, many relative placement tasks cannot be solved perfectly with this constraint. For instance, we might want a point on the handle of a teapot to correspond to some point above a stovetop (which lies outside the convex hull of the points on the stovetop). To allow for such off-object correspondences, we further learn a *residual vector*, $\boldsymbol{\delta}_i^{\mathcal{A}} \in \boldsymbol{\Delta}_{\mathcal{A}}$ for each point $i$ that corrects each virtual corresponding point $\mathbf{v}_i^{\mathcal{A}}$. This allows us to displace each virtual corresponding point to any arbitrary location that might be suitable for the task. To compute these residual vectors, we use a point-wise neural network $g_{\mathcal{R}_{\mathcal{A}}}, g_{\mathcal{R}_{\mathcal{B}}}$ to map each point's embedding into a 3D residual vector:

$$\boldsymbol{\delta}_i^{\mathcal{A}} = g_{\mathcal{R}_{\mathcal{A}}}\left(\boldsymbol{\phi}_i^{\mathcal{A}}\right) \in \mathbb{R}^3, \quad \boldsymbol{\delta}_i^{\mathcal{B}} = g_{\mathcal{R}_{\mathcal{B}}}\left(\boldsymbol{\phi}_i^{\mathcal{B}}\right) \in \mathbb{R}^3$$

Applying these residual offsets to the virtual points, we get a set of *corrected virtual correspondences*, $\tilde{\mathbf{v}}_i^{\mathcal{A}} \in \tilde{\mathbf{V}}_{\mathcal{A}}$ and $\tilde{\mathbf{v}}_i^{\mathcal{B}} \in \tilde{\mathbf{V}}_{\mathcal{B}}$, defined as

$$\tilde{\mathbf{v}}_i^{\mathcal{A}} = \mathbf{v}_i^{\mathcal{A}} + \boldsymbol{\delta}_i^{\mathcal{A}}, \quad \tilde{\mathbf{v}}_i^{\mathcal{B}} = \mathbf{v}_i^{\mathcal{B}} + \boldsymbol{\delta}_i^{\mathcal{B}} \tag{8}$$

These corrected virtual correspondences $\tilde{\mathbf{v}}_i^{\mathcal{A}}$ define the estimated goal location relative to object $\mathcal{B}$ for each point $\mathbf{p}_i \in \mathbf{P}_{\mathcal{A}}$ of object $\mathcal{A}$, and likewise for each point in object $\mathcal{B}$ (see visualization in Appendix A.1).

**Least-Squares Cross-Pose Optimization with Weighted SVD:** Given the sets of dense correspondences, $\left(\mathbf{P}_{\mathcal{A}}, \tilde{\mathbf{V}}_{\mathcal{A}}\right)$ and $\left(\mathbf{P}_{\mathcal{B}}, \tilde{\mathbf{V}}_{\mathcal{B}}\right)$, we would like to compute a single rigid transformation for object $\mathcal{A}$. To do so, we solve for the transformation $\mathbf{T}_{\mathcal{AB}}$ (the cross-pose) that minimizes the weighted distance between each point and its corrected virtual correspondence. Formally, this leads to the following weighted least squares optimization:

$$\mathcal{J}(\mathbf{T}_{\mathcal{AB}}) = \sum_{i=1}^{N_{\mathcal{A}}} \alpha_i^{\mathcal{A}} ||\mathbf{T}_{\mathcal{AB}} \, \mathbf{p}_i^{\mathcal{A}} - \tilde{\mathbf{v}}_i^{\mathcal{A}}||_2^2 + \sum_{i=1}^{N_{\mathcal{B}}} \alpha_i^{\mathcal{B}} ||\mathbf{T}_{\mathcal{AB}}^{-1} \, \mathbf{p}_i^{\mathcal{B}} - \tilde{\mathbf{v}}_i^{\mathcal{B}}||_2^2 \tag{9}$$

where the weights $\alpha_i^{\mathcal{A}} \in \boldsymbol{\alpha}_{\mathcal{A}}$, $\alpha_i^{\mathcal{B}} \in \boldsymbol{\alpha}_{\mathcal{B}}$ signify the importance of each correspondence and are predicted by a point-wise MLP as shown in Figure 4. These weights are learned end-to-end as parameters of our network; they are visualized in Appendix A.2, which shows that the network has learned to assign more weight to the parts of the object that are most important for the task, such as the region around the mug handle (on the mug) and the region around the peg (on the rack). Equation 9 is the well-known weighted Procrustes problem, for which there exists an analytical solution. To maintain the differentiablity of the system, we use a weighted differentiable SVD operation [23] to compute the cross-pose $\mathbf{T}_{\mathcal{AB}}$ that minimizes this objective (see Appendix D for details). This allows us to train the system end-to-end as described below.

## 4.2   TAX-Pose Training Pipeline

To train our model, we use a segmented set of demonstration point clouds of a pair of objects in the goal configuration. For each demonstration point cloud, we generate multiple training examples by transforming each object's point cloud, $\mathbf{P}_{\mathcal{A}}$ and $\mathbf{P}_{\mathcal{B}}$ by random SE(3) transformations $\mathbf{T}_{\alpha}$ and $\mathbf{T}_{\beta}$, respectively. The predicted cross-pose, $\mathbf{T}_{\mathcal{AB}}$, is then compared with the ground truth cross-pose, $\mathbf{T}_{\mathcal{AB}}^{GT} := \mathbf{T}_{\beta}\mathbf{T}_{\alpha}^{-1}$, using an average distance loss [24] with dense regularization (see more details on our training losses in Appendix E.1).

## 5   Experiments

To evaluate TAX-Pose, we conduct a wide range of simulated and real-world experiments on two classes of relative placement tasks: NDF [12] Tasks and PartNet-Mobility Placement Tasks. All tasks involve placing an "action" object at a specific location relative to an anchor object, in which the relative pose is specified by a set of demonstrations. Our method then generalizes to perform this task on novel objects in unseen configurations. We refer the reader to our project website for additional results and videos.

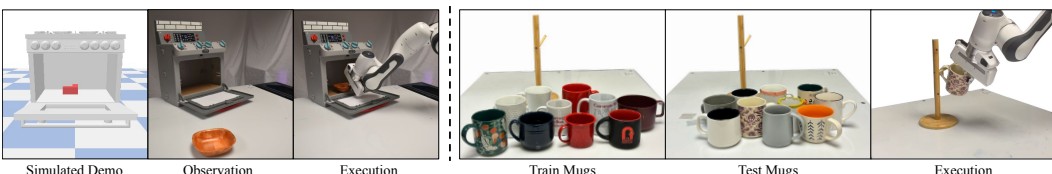

Simulated Demo     Observation     Execution       Train Mugs     Test Mugs     Execution

Figure 5: Real-world experiments summary. **Left:** In object placement task, we train using simulated demonstrations and test on real-world objects. **Right:** Mug Hanging real-world experiments. We train from just 10 demonstrations from 10 training mugs in the real world and test on 10 unseen test mugs.

## 5.1 NDF Tasks

We evaluate our method on all three NDF [12] tasks (*mug* hanging, *bottle* placement, and *bowl* placement); see Appendix F.1.3 for results on *bottle* and *bowl* placement. Results on *mug* hanging are described in more detail below.

**Simulation Experiments:** For our simulation experiments, we perform the task of hanging a mug on a rack as two sequential cross-pose estimation steps: grasping the mug (estimating the cross-pose between the gripper and the mug) and hanging the mug on the rack (estimating the cross-pose between the mug and the rack). In Pybullet [25], we simulate a Franka Panda above a table with 4 depth cameras placed on the corners of the table. The model is trained on 10 simulated demonstrations of mug hanging. We evaluate task execution success on unseen mug instances in randomly generated initial configurations. We measure task success rates of 1) *Grasping*, where success is achieved when the object is grasped stably; 2) *Placing*, where success is achieved when the mug is placed stably on the rack; 3) *Overall*, when the predicted transforms enable both grasp and place success in sequence. We compare our method to Neural Descriptor Field (NDF) [12] and Dense Object Nets (DON) [11]. Details of these methods can be found in prior work [12].

**Simulation Results:** We evaluate our method in simulation in 100 trials consisting of unseen *mug* instances in random initial and goal configurations for both **Upright** and **Arbitrary** poses. As shown in Table 1, our method significantly outperforms the baselines for simulated mug hanging. We report additional results for simulated *bottle* and *bowl* placement tasks in Table 8 in Appendix F.1.3.

**Ablation Analysis:** *Effects of Number of Demonstrations.* To study how the number of demonstrations observed affects our method's performance, we train our model on {10, 5, 1} demonstrations of upright pose mug hanging. Results are found in Table 2. Our method outperforms the baselines for all number of demonstrations; TAX-Pose can perform well even with as few as 5 demonstrations.

*Cross-Pose Estimation Design Choices.* We analyze the effects of design choices made in our Cross-Pose estimation algorithm for the upright pose mug hanging task. Specifically, we analyze the effects of 1) computing residual correspondence; 2) the use of *weighted* SVD over non-weighted in computing cross-pose; 3) using a transformer as the cross-object attention, as opposed to simpler model such as a 3-layer MLP. Table 3 shows that each major component of our system is important for task success. See more ablation experiments in Appendix F.1.1.

|  | Grasp | Place | Overall | Grasp | Place | Overall |
|---|---|---|---|---|---|---|
|  | **Upright Pose** | | | **Arbitrary Pose** | | |
| DON [11] | 0.91 | 0.50 | 0.45 | 0.35 | 0.45 | 0.17 |
| NDF [12] | 0.96 | 0.92 | 0.88 | **0.78** | 0.75 | 0.58 |
| **TAX-Pose** | **0.99** | **0.97** | **0.96** | 0.75 | **0.84** | **0.63** |

Table 1: Mug on rack simulation success rate (↑)

| Model | # Demos Used | | |
|---|---|---|---|
|  | 1 | 5 | 10 |
| DON [11] | 0.32 | 0.36 | 0.45 |
| NDF [12] | 0.46 | 0.70 | 0.88 |
| **TAX-Pose** | **0.77** | **0.90** | **0.96** |

Table 2: # Demos vs. Overall success rate (↑)

| Ablation | Grasp | Place | Overall |
|---|---|---|---|
| No Res. | 0.97 | 0.96 | 0.93 |
| Unw. SVD | 0.92 | 0.94 | 0.88 |
| No Attn. | 0.90 | 0.82 | 0.76 |
| **TAX-Pose** | **0.99** | **0.97** | **0.96** |

Table 3: Mug hanging ablations success rate (↑)

**Real-World Experiments:** We explore the hanging component of the mug on a rack task in a real world environment, which requires estimating the cross-pose between the mug and the rack. We train TAX-Pose using real demonstrations of 10 different mugs hung on a rack (1 demonstration each, for a total of only 10 real-world demonstrations for training). A motion primitive is used to grasp each mug, after which the robot plans a trajectory to apply the predicted cross-pose to the grasped mug. We evaluate the model on the 10 training mugs in novel poses, as well as on 10 unseen mugs (see Figure 5). For each of the 20 mugs, we conduct 5 trials, varying the mug's and rack's starting poses in each trial. Success is recorded if a peg penetrates the mug handle at the end of the trial. Our method achieves a success rate of 62% on training mugs in novel poses and 54%

on unseen mugs. A visualization of the results can be seen in Figure 5 (right) and on the project website. Note that our method is able to perform the mug hanging task while varying the pose of the mug rack (see our project website), whereas the baselines (NDF [12], DON [11]) cannot because they assume a fixed, known rack position (see NDF [12] for baseline details).

## 5.2 PartNet-Mobility Placement Tasks

**Task Description**: We also define a PartNet-Mobility Placement task as placing a given action object relative to an anchor object based on a semantic goal position. We select a set of household furniture objects from the PartNet-Mobility dataset [26] as the anchor objects, and a set of small rigid objects released with the Ravens simulation environment [27] as the action objects. For each anchor object, we define a set of semantic goal positions (i.e. 'top', 'left', 'right', 'in'), where action objects should be placed relative to each anchor. Each semantic goal position defines a unique task in our cross-pose prediction framework. Given a synthetic point cloud observation of both objects, the task is to predict a cross-pose that places the object at the specific semantic goal. We evaluate both a *goal-conditioned* variant (**TAX-Pose GC**), which is trained across all goals, and a *task-specific* variant (**TAX-Pose**) of our model, which trains a separate model per goal type (see Appendix F.2.2 for details). In both cases we train only 1 model across all action and anchor objects. All models are trained entirely on simulated data and transfer directly to real-world with no finetuning. Further task details can be found in the Appendix G.2.

**Baselines**: We compare our method to a variety of end-to-end imitation-learning-based methods trained from a motion planner expert in simulation (see Appendix G.2.4 for details). Note that in the PartNet-Mobility Placement experiments, the pose of the anchor object poses are randomly varied. As such, we omit a comparison to methods that assume a static anchor, such as the Neural Descriptor Field (NDF) [12] and Dense Object Nets (DON) [11] baselines used in the mug hanging task (Section 5.1), as both methods assume that the anchor objects are in a fixed, known position.

**Results**: We report rotation ($\mathcal{E}_\mathbf{R}$) and translation ($\mathcal{E}_\mathbf{t}$) error between our predicted transform and the ground truth as geodesic rotational distance [28, 29] and $L2$ distance, respectively. In both our simulated experiments (Table 4 Top) and our real-world experiments (Table 4 Bottom), we find that TAX-Pose outperforms the baseline end-to-end imitation learning methods, with the *goal-conditioned* variant, TAX-Pose GC, performing the best. In real-world experiments, our method generalizes to novel distributions of starting poses better than the Goal Flow baseline, placing action objects into the goal regions with a 92% success rate. See Figure 5 (left) and the website for results; see Appendix F.2 for more detailed tables and Appendix G.2.4 for baseline details.

|  | Average | |
|---|---|---|
|  | $\mathcal{E}_\mathbf{R}$ | $\mathcal{E}_\mathbf{t}$ |
| E2E BC | 42.26 | 0.73 |
| E2E DAgger | 37.96 | 0.69 |
| Traj. Flow | 35.95 | 0.67 |
| Goal Flow | 26.64 | 0.17 |
| TAX-Pose | 6.64 | **0.16** |
| **TAX-Pose GC** | **4.94** | **0.16** |

|  | Average SR |
|---|---|
| Goal Flow | 0.31 |
| **TAX-Pose** | **0.92** |

Table 4: **Top**: Simulation Rotational (°) and Translational (m) Errors (↓). **Bottom**: Real-world goal placement success rate (↑).

## 6 Conclusion and Limitations

In this paper, we show that dense soft correspondence can be used to learn task specific object relationships that generalize to novel object instances. Correspondence residuals allow our method to estimate correspondences to virtual points, outside of the objects convex hull, drastically increasing the number of tasks this method can complete. We further show that this "cross-pose" can be learned for a task, using a small number of demonstrations. Finally, we show that our method far outperforms the baselines on two challenging tasks in both real and simulated experiments. While our method is able to predict relative pose relationships with high precision, it has several limitations:

- **Requires segmentation**: Our method requires an accurate segmentation of two objects in order to predict their relative goal pose.

- **Performance degrades under occlusion**: Our method performs best when complete point clouds are provided, captured via multiple cameras or by repeatedly reorienting the objects.

- **Poorly defined for multimodal relationships**: Because our method extracts a single global estimate of relative pose from a fixed set of correspondences, performance on objects with multiple valid goals is not well-defined. Our method might be augmented with a consensus-based or sampling-based approach to capture the multimodality of the solution space in these cases. We leave this for future work.

## Acknowledgements

This material is based upon work supported by the National Science Foundation under Grant No. IIS-1849154. This work was also supported by LG Electronics. We are grateful to Daniel Seita and Jenny Wang for their helpful feedback and discussion on the paper.

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
