# OpenReview forum: "TAX-Pose: Task-Specific Cross-Pose Estimation for Robot Manipulation"
_robot-learning.org/CoRL/2022/Conference — CoRL 2022 Poster_

### Official Review · Reviewer_r914 · 2022-07-12

**Originality:** Very Good
**Technical Quality:** Excellent
**Clarity Of Presentation:** Very Good
**Impact:** 4

**Recommendation:**

Strong Accept: I recommend accepting the paper and will argue for my recommendation even if other reviewers hold a different opinion.

**Summary:**

The paper proposes an approach to estimate the appropriate transformation to apply to an object in order to achieve the desired relative pose between two objects in a given task. Experimental evaluation shows convincing results in two different settings.

**Issues:**

1. Can the authors comment on if it is possible to extend the framework to tasks that involve more than two objects?
2. I wonder if the authors have thought about using some SE(3)-equivariant network architecture like the vector neuron used in NDF [34]. Will that simplify the training process?
3. There are some minor notation issues:
    1. Should $\tilde{\mathbf{V}}_\mathcal{A}$ and $\tilde{\mathbf{V}}_\mathcal{B}$ in Equation 6 and 7 be swapped? If I understand line 167 and 168 correctly, $\tilde{\mathbf{V}}_\mathcal{A}$ is the correspondence for  $\mathbf{P}_\mathcal{B}$. In other words, $\mathbf{P}_\mathcal{A} \in \mathbb{R}^{3\times N_A}$ , then $T^{GT}_\mathcal{AB}\mathbf{P}_\mathcal{A}\in \mathbb{R}^{3\times N_A}$. However, based on the definition of $\tilde{\mathbf{V}}_\mathcal{A}$ in line 167 and 168, $\tilde{\mathbf{V}}_\mathcal{A} \in \mathbb{R}^{3\times N_B}$.
    2. What are $\tilde{\mathbf{P}}_\mathcal{A}$ and $\tilde{\mathbf{P}}_\mathcal{B}$ in Figure 3? Those symbols does not exist in the paper.

**Quality Of The Limitations Section:**

Additional details required

**Reviewer Expertise:**

4: The reviewer is confident but not absolutely certain that the evaluation is correct

**Robotics Focus:**

Sufficient demonstration on hardware

**Strengths And Weaknesses:**

Strengths:

1. The paper studies an interesting perspective in manipulation, treating a manipulation task as achieving a desired relative pose between objects. The proposed method is novel.
2. The paper is well-written (despite some minor notation issues) and technically sound.
3. The experimental results are compelling.

Weaknesses:

1. The framework seems hard to extend to tasks involving more than two objects. What if a task requires five objects to achieve some desired relative poses?
2. There are some minor notation issues (see the ‘Issues’ section below).

**Summary Of Recommendation:**

The paper studies an important problem in manipulation, defining the task as achieving desired relative pose between objects, and proposes a novel framework to solve it. The paper is well-organized. Although it seems like the framework is constrained in a two-object scenario, I believe the paper is a strong contribution and can potentially have a major impact.

---

> ### Author Response · Authors · 2022-08-27
> **Response to Reviewer r914**
>
> **Comment:**
>
> Thank you for the valuable feedback and suggestions! The raised questions are addressed as follows.
>
> Revised text and tables are colored in magenta in the newly revised paper and supplement attached.
>
> **Q: “1. The framework seems hard to extend to tasks involving more than two objects. What if a task requires five objects to achieve some desired relative poses?”**
>
> **A:** While the proposed method is designed to handle relative relationships between pairs of objects, this is in fact a modular formulation that can be relatively easily extended to some tasks involving multiple objects by leveraging the TAX-Pose framework as a modular component in a multi-object task system. We can formulate multi-object tasks as a composition of object-object pair tasks and consider the TAX-Pose between each object pair. For example, the mug hanging task in our experiment actually involved three objects, the gripper, the mug, and the rack. Following prior work [1], we break this into 2 subtasks, involving the pairwise interaction between the gripper and the mug, and then the pairwise interaction between the mug and the rack.
>
> Further, we believe that there is a range of tasks that can be described by pairwise interactions, such as placing a lasagna in an oven, place a pot on a stove, placing a plate in a microwave, place a mug onto a mug rack, placing a lid on a jar, or placing a cup onto a shelf. Some more complex tasks can be broken down into a sequence of pairwise tasks as described above.  Although there are other tasks that cannot be described this way, we believe that this set of tasks is still sufficiently large and important to be interesting and worth studying.
>
> [1] Simeonov, Anthony, et al. "Neural descriptor fields: Se (3)-equivariant object representations for manipulation." 2022 International Conference on Robotics and Automation (ICRA). IEEE, 2022.
>
> **Q: “I wonder if the authors have thought about using some SE(3)-equivariant network architecture like the vector neuron used in NDF [34]. Will that simplify the training process?”**
>
> **A:** Thank you for the suggestion. In fact we briefly experimented with using a Vector Neuron network as the feature extractor network in place of the currently used DGCNN; however, we found that in fact it led to worse task performance, as briefly mentioned in Section 4.2 of the main text. We are unsure why this might be the case; one possibility is that the equivariances in Vector Neurons limit the expressivity of the network; specifically, there is limited mixing of information across the object dimensions in Vector Neurons, outside of the nonlinearities. Additionally, since our method needs to learn rotational invariance, our network might learn a better part level featureiztion that aids in generalization across objects in a given class. We leave a deeper analysis of Vector Neuron networks to future work.
>
> *We have also updated the raised notations in the revised text of the paper. We greatly appreciate your feedback, and please let us know if these changes have addressed your concerns and if there are other questions you may have!*
>
>
> **Zip File:**
>
> /attachment/46431f011e90752f6e6f3f8b388da1594fab9541.zip

---

### Official Review · Reviewer_d5cK · 2022-07-27

**Originality:** Good
**Technical Quality:** Very Good
**Clarity Of Presentation:** Very Good
**Impact:** 3

**Recommendation:**

Weak Accept: I recommend accepting the paper, but will not argue for my recommendation if the majority of other reviewers have a different opinion.

**Summary:**

This paper tackles the problem of estimating the relative pose between two objects conditioned to a specific manipulation task. This pose is then used to guide a robot manipulator to pick and place one object in a desired location relative to the other.
The authors given a formal definition of the estimation task by introducing the concept of "cross-pose".
The proposed system, called TAX-Pose, is built upon Deep Closest Point (DCP)[20], with the addition of specific components required by the manipulation task, and OMPL for motion planning.
The different components of the system are validate in ablation studies.
Experiments show that TAX-Pose achieves state-of-the-art performance and it is able to generalize to different objects of the same category.
Sim-to-real transfer is achieved without fine-tuning on some of the manipulation tasks tested in simulation. The proposed system is able to achieve much higher success rate than the baseline in the real-world.



**Issues:**

Besides the comments regarding weaknesses in the corresponding section, I include some other minor issues below:

- The introduction of the variables $T_{\alpha}$ and $T_{\beta}$ in the section "Problem Statement" seems not necessary since such variables are not used anymore in the text.

- It seems that \tilde{P}_{A} in Fig. 3 is call \tilde{V}_{A} in the text. Same for B.

- Reference [34] and [35] cite the ArXiv version of the papers. However, the works have been published in ICRA 22 and CoRL 18 respectively. I would suggest to update these references.

**Quality Of The Limitations Section:**

Additional details required

**Reviewer Expertise:**

3: The reviewer is fairly confident that the evaluation is correct

**Robotics Focus:**

Sufficient demonstration on hardware

**Strengths And Weaknesses:**

-- Strengths

- The paper is well-written and easy to read. The authors were able to describe the problem statement and methodology in a clear and descriptive way in spite of the little space available.  The paper contains all the information and references that the reader needs to understand the proposed method and make connections to the current state-of-the-art.

- The definition of the term of "cross-pose" is intuitive and its mathematical formulation is sound. The reader needs to check the Appendix to get the full idea behind this definition.

- The proposed system, TAX-Pose, is a slightly adapted version of DCP for the task of "cross-pose" estimation. The authors propose to add a residual net block, which applies a residual vector to each point correspondence, to account for the case when the object are not in contact or overlap. I find this an elegant solution.

- The experiments are relevant. The different components of the system are validated in ablation studies. TAX-Pose achieves state-of-the-art results in simulation. Simulation to real-world transfer is achieved without fine-tuning, thanks to, according to the authors, the translation-equivariant property of TAX-Pose.

-- Weaknesses

- The originality of this work is a weakness. TAX-Pose heavily builds upon DCP for the relative pose estimation and completely rely on OMPL for planning. The main difference with respect to DCP is a revised version of the block that in DCP is called "Pointer". In this work a residual net is used to account for the case when the object are not in contact or overlap. Although this is an elegant solution, it is a minor increment with respect to the DCP architecture.

- I would suggest  to give more insights about the results. In particular:
Do the authors have an intuition why the rotation results of TAX-Pose are much better that the baselines?
Why the sim-to-real transfer capabilities of TAX-Pose are much better than Goal-Flow? I would expect that the translation-equivariant would improve the sim-to-real transfer capabilities of TAX-Pose but the results of Tab. 2 show a big gap with respect to Goal-Flow. Could the authors explain their intuition about the reason behind this result?
Why was it not possible to evaluate the mug hanging experiments in the real-world?


**Summary Of Recommendation:**

This paper has some potentials to have an impact in the field.
Although the methodology is an incremental contribution with respect to DCP, the authors have shown that DCP can be used to solve a relative pose problem rather than point-cloud registration problem.
The experiments are relevant and the ablation studies useful to validate the different components of the system. However, further discussions on the results would improve the quality of the work.

**After Rebuttal**
I appreciate the authors' response to my concerns. In my opinion, this work is a good contribution for the conference. I confirm my initial recommendation of weak accept.

---

> ### Author Response · Authors · 2022-08-27
> **Response to Reviewer d5cK**
>
> **Comment:**
>
> Thank you for the valuable feedback and suggestions! The raised questions are addressed as follows.
>
> Revised text and tables are colored in magenta in the newly revised paper and supplement attached.
>
> **Q: “The originality of this work is a weakness. TAX-Pose heavily builds upon DCP for the relative pose estimation and completely rely on OMPL for planning. The main difference with respect to DCP is a revised version of the block that in DCP is called "Pointer". In this work a residual net is used to account for the case when the object are not in contact or overlap. Although this is an elegant solution, it is a minor increment with respect to the DCP architecture.”**
>
> **A:** While this work is built upon DCP, the task setting is quite different from the original DCP algorithm. DCP focused on point-cloud registration (aligning one view of an object to another view of the *same* object). One of our innovations was realizing that DCP can be applied to relative placement tasks by learning to compute the cross-pose (a novel formulation that we proposed in this work) between two *different* objects. This insight allowed us to significantly outperform prior work (NDF, DON) on relative placement tasks, which suggests that the idea in our work is non-obvious and effective. Further, prior methods “assumes a static environment that remains fixed between demonstration time and test-time” [22]. Unlike those prior methods, our method can generalize to a non-fixed anchor object, i.e. the anchor object pose does not need to be explicitly estimated in our method; we instead estimate the cross-pose between a pair of action and anchor object directly. We have updated the Related Work section to further emphasize the benefits of our method as compared to prior work.
>
> From a technical standpoint, our work is the first to introduce the notion of cross-pose, which we define in Section 3 and analyze theoretically in Supplement Section 1. Extending the framework from DCP, we learn a residual to the soft correspondences, as the reviewer noted, allowing for points to match outside the convex hull of each object. This component is necessary when matching between objects of drastically different morphologies. Also, DCP does not use a weighted SVD; in contrast, we learn importance weights for different regions of the point cloud, allowing the system to focus on certain regions of each object that are important for the given task, which we integrate into a weighted differentiable SVD. The ablations in Table 5 of the paper show that each of these components contribute to our final performance. We also prove that our method has translational equivariance in Supplement Section 2, which was not done in prior work.
>
> Thus, our paper contains a number of new technical insights and novelties that are not contained in prior work.
>
> **Q: “Do the authors have an intuition why the rotation results of TAX-Pose are much better that the baselines? Why the sim-to-real transfer capabilities of TAX-Pose are much better than Goal-Flow? I would expect that the translation-equivariant would improve the sim-to-real transfer capabilities of TAX-Pose but the results of Tab. 2 show a big gap with respect to Goal-Flow."Could the authors explain their intuition about the reason behind this result?”**
>
> **A:** Our intuition is that, by defining the problem as learning a set of weighted correspondences, our method learns to focus on the important relationships between the objects for the task. Our attention mechanism also allows our method to generalize to novel objects, by focusing on the task-relevant object features and relationships. As the reviewer noted, our method also largely benefits from translation equivariance, as shown in Supplement Section 2. These components contribute to our success in sim2real transfer.
>
> *We greatly appreciate your feedback, and please let us know if these changes have addressed your concerns and if there are other questions you may have!*
>
>
> **Zip File:**
>
> /attachment/cb5f035716b742d6777b0378197fda3998fbbe9c.zip

---

### Official Review · Reviewer_4CrR · 2022-07-28

**Originality:** Fair
**Technical Quality:** Good
**Clarity Of Presentation:** Good
**Impact:** 2

**Recommendation:**

Weak Reject: I recommend rejecting the paper, but will not argue for my recommendation if the majority of other reviewers have a different opinion.

**Summary:**

The paper proposed a vision system that uses pose relationships between two objects for robot manipulation. Point clouds of two interacting objects are retrieved through DGCNN and the proposed TAX-Pose method learns to predict a task-specific pose transformation from demonstrations. The prediction is then used to plan a trajectory that accomplishes the task. The system has been tested in both simulation and real-world environments and shown that the model sample efficiency that can learn from a small number of demonstrations. Dense soft correspondence + residuals allows the method to generalize to novel object instances.

**Issues:**

See above

**Quality Of The Limitations Section:**

Limitations are addressed clearly

**Reviewer Expertise:**

4: The reviewer is confident but not absolutely certain that the evaluation is correct

**Robotics Focus:**

Sufficient demonstration on hardware

**Strengths And Weaknesses:**

The author designed a cross-pose estimation system that helped with object manipulation. Comprehensive experiments were provided to show that the proposed method is beneficial in both simulation and real world.
The downside of the paper is that the proposed method injects a significant amount of inductive biases to solve pick-and-place-ish tasks. The scalability of the method is questionable. First the method is heavily rely on a relatively good pretrained model for point cloud extraction e.g. DGCNN as used in the paper. It turns the problem setup from image space to feature space that makes the claim of a “vision-based system” a bit weak. In addition, the feature of zero-shot sim to real transfer and generalization to novel objects of the same category likely comes from the use of DGCNN. Secondly, the scalability is significantly limited once a model is trained. It only works for the exact pair of objects and for one specific task. It is not so clear by reading the paper, but it seems a model trained for “Left” task would not be able to do “Right” task. Or a model trained for “cube-left-oven” might not be able to do “cube-left-microwave”. Thirdly, the proposed method works only for pairs of objects. It seems hard to scale the system to support multiple objects or even generalize to the number of objects. In practice, real-world tasks often involve multiple objects and the complex dynamic among objects interaction.

As the author mentioned in the limitation section, symmetry is not handled. However, Rotational errors (ER) are reported in the evaluation metric, which might not make much sense when objects like cubes, bottles, plates etc are involved.


**Summary Of Recommendation:**

The scalability of the method is questionable and is seems the model can only operate in a very constrained scenario. As symmetry is not haddled, the comparision between TAX-Pose and baseline methods are a bit weak. Rotational errors (ER) is no longer a fair metric. In terms of the translational errors (Et), TAX-Pose does not show much advantages.

---

> ### Author Response · Authors · 2022-08-27
> **Response to Reviewer 4CrR (1/3)**
>
> **Comment:**
>
> Thank you for the valuable feedback and suggestions! The raised questions are addressed as follows.
>
> Revised text and tables are colored in magenta in the newly revised paper and supplement attached.
>
> **Q: “The downside of the paper is that the proposed method injects a significant amount of inductive biases to solve pick-and-place-ish tasks.”**
>
> **A:** Most previous methods that do not inject inductive bias require a large number of demonstrations, an accurate simulator, and/or a large number of real-world interactions. In contrast, our method is able to learn to generalize to novel objects within a category from just 5 demonstrations with 90% success, and from 10 demonstrations with 96% success (Table 4).  We achieve this by using an architecture that reasons about the cross-pose between a pair of objects.  Although there is some inductive bias in our method, it is still general enough to solve a range of placement tasks, including placing an object in/on/next to an anchor object (PartNet-Mobility tasks), placing a gripper on a mug rim, or placing a mug handle on a mug rack, without any changes in the method or architecture. Moreover, many object rearrangement tasks can arguably be reduced to a series of pick-and-place tasks, each of which could incorporate our cross-pose prediction module to predict intermediate goal states for an agent to reach. Thus, while our method does not solve all robot manipulation tasks, it is a significant advance on a set of tasks that is broad and important and hence worthy of study.
>
>
> **Q: “The scalability of the method is questionable. First the method is heavily rely on a relatively good pretrained model for point cloud extraction e.g. DGCNN as used in the paper. It turns the problem setup from image space to feature space that makes the claim of a “vision-based system” a bit weak. In addition, the feature of zero-shot sim to real transfer and generalization to novel objects of the same category likely comes from the use of DGCNN.”**
>
> **A:** The only pretraining used is our own procedure to learn rotationally invariant embeddings, mentioned at the end of Section 4.2 in the paper and in Supplement Section 5.1  Before this pre-training, the network is randomly initialized. We have updated Supplement Section 5.1 of the paper to clarify this point.  Additionally, all of the baselines use a similar version of pretraining, so our method does not get any additional benefit from this pre-training compared to the baselines that we significantly outperform.
>
> Further, we performed an additional experiment in which we removed the pre-training from our method but trained for 5 times longer; we find that such an approach achieves slightly worse results as with the pretraining; this can be seen in the results added in Table S2 in Supplement Section 4.2 (with pre-training: 96% success; without pre-training: 92% success). Thus pre-training leads to slightly better performance; another benefit is that the pre-training is done in a multi-task way, so the network can be more quickly adapted to new tasks after the pre-training is performed. For example, we use the same pre-trained mug embeddings for both the gripper-mug cross-pose estimation for grasping as well as the mug-rack cross-pose estimation for mug hanging. We have clarified this point in Supplement Section 4.2.
>
>
> **Zip File:**
>
> /attachment/2f0b0c2ea1eeef846950c5a76ff1b61a0abb4a4d.zip

---

> > ### Author Response · Authors · 2022-08-27
> > **Response to Reviewer 4CrR (2/3)**
> >
> > **Q: “Secondly, the scalability is significantly limited once a model is trained. It only works for the exact pair of objects and for one specific task. It is not so clear by reading the paper, but it seems a model trained for “Left” task would not be able to do “Right” task. Or a model trained for “cube-left-oven” might not be able to do “cube-left-microwave.”**
> >
> > **A:** We acknowledge the potential confusion caused in the paper. To clarify, for the PartNet-Mobility objects placement task, we train a single model across all anchor/action categories, one model per semantic task (for a total of 4 models). This means that the “cube-left-oven” and “bowl-left-microwave” tasks are predicted by the same network, without being explicitly told that the anchor is an oven or a microwave. We have updated the text in Section 5.1 of the paper to make this more clear.
> >
> > Furthermore, inspired by feedback from several reviewers, we have designed a goal-conditioned version of our policy which allows us to train a single model to perform placement across different semantic placement locations. This requires just a small modification to our method: we assign each of the semantic tasks (top, in, left, right, …) a unique label, which we encode into a latent space with a single linear layer. Then, we incorporate this latent context vector in the same manner as in the original DGCNN [33] paper, by concatenating it to the final representation layer of the DGCNN backbone. We train this network on the PartNet-Mobility dataset across all semantic tasks, providing the appropriate semantic label as an input. With this procedure, we are able to use just a single network across all semantic task variations. We find that this simple modification to our network and training procedure produces models which perform nearly as well as semantic task-specific models, with respect to both rotational and translational error:
> >
> > | **Method** | **Avg Angular Error (degree)** | **Avg. Translational Error (m)** |
> > | :-: | :-: | :-: |
> > | TAX-Pose (original) | 6.64 | 0.16 |
> > | TAX-Pose (goal-conditioned) | 7.74 | 0.17 |
> >
> > See Section 5.1, Table 1 of the new manuscript for full results.  The goal-conditioned training procedure is described in the “Overall Training Procedure” paragraph of Section 4.2.
> >
> > **Q: “Thirdly, the proposed method works only for pairs of objects. It seems hard to scale the system to support multiple objects or even generalize to the number of objects. In practice, real-world tasks often involve multiple objects and the complex dynamic among object interactions.”**
> >
> > **A:** While the proposed method is designed to handle relative relationships between pairs of objects, this is in fact a modular formulation that can be relatively easily extended to some tasks involving multiple objects by leveraging the TAX-Pose framework as a modular component in a multi-object task system. We can formulate multi-object tasks as a composition of object-object pair tasks and consider the TAX-Pose between each object pair. For example, the mug hanging task in our experiment actually involved three objects, the gripper, the mug, and the rack. Following prior work [1], we break this into 2 subtasks, involving the pairwise interaction between the gripper and the mug, and then the pairwise interaction between the mug and the rack.
> >
> > Further, we believe that there is a range of tasks that can be described by pairwise interactions, such as placing a lasagna in an oven, place a pot on a stove, placing a plate in a microwave, place a mug onto a mug rack, placing a lid on a jar, or placing a cup onto a shelf. Some more complex tasks can be broken down into a sequence of pairwise tasks as described above.  Although there are other tasks that cannot be described this way, we believe that this set of tasks is still sufficiently large and important to be interesting and worth studying.
> >
> > [1] Simeonov, Anthony, et al. "Neural descriptor fields: Se (3)-equivariant object representations for manipulation." 2022 International Conference on Robotics and Automation (ICRA). IEEE, 2022.

---

> > > ### Author Response · Authors · 2022-08-27
> > > **Response to Reviewer 4CrR (3/3)**
> > >
> > > **Q: “As the author mentioned in the limitation section, symmetry is not handled. However, Rotational errors (ER) are reported in the evaluation metric, which might not make much sense when objects like cubes, bottles, plates etc are involved.”**
> > >
> > > **A:** The reviewer is correct that the evaluation metric does not fully capture the intuitive notion of task success for some placement tasks in which there might be a set of poses that are all equally valid.  However, we note that the evaluation metric is consistent for all baselines.  Further, some of the objects that we evaluate on are not symmetric, such as mugs. In the PartNet-Mobility objects placement task, the L-shape block and the toy part object are also asymmetric.  Training an estimator to predict the full set of possible valid cross-poses in a symmetry-aware manner is an interesting direction of future work. However, substantial changes to the method AND the datasets would be required - the NDF and PartNet-Mobility datasets only define a single “canonical” goal pose for each pair of objects and would need to be augmented with samples from the full goal sets for each object (inclusive of object symmetry). Still, we believe that our method is an important advance on relative placement tasks.
> > >
> > > *We greatly appreciate your feedback, and please let us know if these changes have addressed your concerns and if there are other questions you may have!*

---

### Official Review · Reviewer_gUTd · 2022-07-31

**Originality:** Good
**Technical Quality:** Good
**Clarity Of Presentation:** Fair
**Impact:** 3

**Recommendation:**

Weak Reject: I recommend rejecting the paper, but will not argue for my recommendation if the majority of other reviewers have a different opinion.

**Summary:**

The paper introduces the notion of “cross-pose”, a task-specific relationship between objects that captures a desired goal configuration. The proposed “cross-pose” function maps pairs of object pointclouds to a relative pose in SE(3); the function value is defined to be identity when the objects satisfy the relative pose goal relationship. The model is trained on a dataset of demonstrations of the goal condition, e.g. a mug hung on a rack, and is designed to operate at the category level for “action” and “anchor” objects.

The main contribution of the paper is the formulation of cross-pose and an architecture for learning it. The central novelty in the problem formulation (in particular relative to Neural Descriptor Fields) is that both the action and anchor object are encoded, enabling generalization at the category level for both (in contrast to NDF, in which only the “action” object is embedded).

**Issues:**

Please see above recommendations regarding the related work and result sections.

I would recommend moving the success rate tables to the main paper - the error in rotation/translation is difficult to assess task performance with.

Question regarding the partnet simulation experiment: is the anchor object pose randomized, or just the target object? The description in 6.1 is ambiguous.

The hardware results describe at length the modifications required to achieve the real world experiments, though the mug-hanging experiments demonstrate that the method is able to learn relative grasp poses. Is there a particular reason why this was forgone for the hardware experiment?

Minor typos:

Appendix 7.1 "quarry"
Section 5.1 “a semantic goal positions”

**Quality Of The Limitations Section:**

Additional details required

**Reviewer Expertise:**

3: The reviewer is fairly confident that the evaluation is correct

**Robotics Focus:**

Sufficient demonstration on hardware

**Strengths And Weaknesses:**

The paper studies an important problem. The description and analysis of the cross-pose function are thorough. The paper achieves impressive results on the simulated mug-hanging task in comparison to a recent state-of-the-art system, and the ablation results motivate the design decisions.

The paper has some limitations which the authors are transparent about - the strongest assumption is that there is one such “cross-pose” which satisfies the relationship. In reality, there are a manifold of poses which satisfy the goal condition for these tasks, though this is a reasonable first approximation inherited from previous work. A larger conceptual limitation is the restriction of tasks to pick and place. While a large number of household tasks may be specified by a static observation, many cannot - e.g. “carry this mug of tea” is not well specified this way. Another limitation present from the evaluation is the need to train one separate semantic model for each variant of a task - i.e. each semantic placement (right, left, in, …) requires a unique model to be trained. We’ve seen methods like CLIPort and Socratic Models use natural language in conjunction with vision to handle such variation. This isn’t the focus of this work but it warrants mention.

The related work section is relatively weak. In particular, it fails to mention the baselines compared to in the mug hanging experiment section, the most similar related work, as well as related ideas such as keypoint methods or canonicalization (e.g. NOCS). From the text alone, the novelty of the proposed method relative to these baselines is not clearly conveyed.

The comparison to existing methods is also a bit lacking. The related methods are absent from the PartNet evaluation without mention. I believe they are omitted because the anchor object pose is also varied, where DON and NDF would require a fixed anchor. This at least warrants clarification in the text (if not outright comparison in a "fixed anchor" experiment). Moreover, only one object category experiment from NDF was replicated - it's difficult to assess the improvement in performance offered by the proposed approach with a limited evaluation.

**Summary Of Recommendation:**

I'm recommending weak rejection on the basis of the paper needing substantial improvements to the quality of presentation and potentially additional experimentation. At the moment, the novelty is not clear relative to existing baselines and the related work, experiment setup, and results lack sufficient discussion.

---

> ### Author Response · Authors · 2022-08-27
> **Response to Reviewer gUTd (1/2)**
>
> **Comment:**
>
> Thank you for the valuable feedback and suggestions! The raised questions are addressed as follows.
>
> Revised text and tables are colored in magenta in the newly revised paper and supplement attached.
>
> **Q: “The paper has some limitations which the authors are transparent about- the strongest assumption is that there is one such “cross-pose” which satisfies the relationship. In reality, there are a manifold of poses which satisfy the goal condition for these tasks, though this is a reasonable first approximation inherited from previous work.”**
>
> **A**: We agree that our method currently is limited to unambiguous goal-placement tasks with only a single valid placement pose (an assumption inherited from previous work, as the reviewer noted). We have clarified in Section 3 in the updated paper that some tasks admit a set of solutions (e.g. when there are object symmetries, or when the goal is semantically defined, such as “place the object somewhere on top of the table”) but that we focus on cases in which “there is a single, unambiguous relative pose needed to achieve a given task.”  We also mention this in our limitations section, as the reviewer has noted. That said, there are a large number of placement tasks for which there is a single unambiguous placement pose (e.g. inserting an asymmetric peg into a hole, precisely arranging asymmetric objects in a scene, or stacking asymmetric objects so that they are in perfect alignment); therefore we believe that our method is an important advance on relative placement tasks.
>
> When there is a larger goal set in a placement task, training an estimator to predict the full set of possible valid cross-poses is an interesting direction of future work. However, substantial changes to the method AND the datasets would be required - the NDF and PartNet-Mobility datasets only define a single “canonical” goal pose for each pair of objects and would need to be augmented with samples from the full goal sets for each object (inclusive of object symmetry or semantic ambiguity in goal location). Still, we believe that our method is an important advance on relative placement tasks.
>
> **Q: “Restriction of tasks to pick and place. While a large number of household tasks may be specified by a static observation, many cannot - e.g. “carry this mug of tea” is not well specified this way.”**
>
> **A:** Our work currently focuses on goal state prediction, or predicting “where” a specified object should end up to achieve task success. This prediction task is mostly independent of “how” an agent should achieve this goal (e.g. carry the mug of tea upright without spilling), although this is indeed a necessary component of accomplishing many tasks. For simple pick-and-place tasks, given an observed start and predicted goal, we can compute the trajectory with a simple motion planner to move the object to the goal - these are the cases we consider in this work. In more complex scenarios, we might instead use a constrained planner or learned goal-conditioned policy to do so. But in all cases, predicting the desired goal position of an object - the focus of this work - is necessary to compute.
>
>
> **Zip File:**
>
> /attachment/e7e9201e7f8bdf8cf29ee12928eb2921e90ccc78.zip

---

> > ### Author Response · Authors · 2022-08-27
> > **Response to Reviewer gUTd (2/2)**
> >
> > **Q: “The need to train one separate semantic model for each variant of a task - i.e. each semantic placement (right, left, in, …) requires a unique model to be trained. We’ve seen methods like CLIPort and Socratic Models use natural language in conjunction with vision to handle such variation. This isn’t the focus of this work but it warrants mention.”**
> >
> > **A:** Inspired by this feedback, we have designed a goal-conditioned version of our policy which allows us to train a single model to perform placement across different semantic placement locations. This requires just a small modification to our method: we assign each of the semantic tasks (top, in, left, right, …) a unique label, which we encode into a latent space with a single linear layer. Then, we incorporate this latent context vector in the same manner as in the original DGCNN [33] paper, by concatenating it to the final representation layer of the DGCNN backbone. We train this network on the PartNet-Mobility dataset across all semantic tasks, providing the appropriate semantic label as an input. With this procedure, we are able to use just a single network across all semantic task variations. We find that this simple modification to our network and training procedure produces models which perform nearly as well as semantic task-specific models, with respect to both rotational and translational error:
> >
> > | **Method** | **Avg Angular Error (degree)** | **Avg. Translational Error (m)** |
> > | :-: | :-: | :-: |
> > | TAX-Pose (original) | 6.64 | 0.16 |
> > | TAX-Pose (goal-conditioned) | 7.74 | 0.17 |
> >
> > See Section 5.1, Table 1 of the new paper manuscript for full results. The goal-conditioned training procedure is described in the “Overall Training Procedure” paragraph of Section 4.2 in the updated paper.
> >
> > We agree that language-conditioned models are outside the scope of this work, but worth mentioning. Therefore we also have added a description of language conditioned policies, such as CLIPort and Socratic Models, to our related work section.
> >
> > **Q: “The related methods are absent from the PartNet evaluation without mention. I believe they are omitted because the anchor object pose is also varied, where DON and NDF would require a fixed anchor. This at least warrants clarification in the text (if not outright comparison in a "fixed anchor" experiment). Moreover, only one object category experiment from NDF was replicated - it's difficult to assess the improvement in performance offered by the proposed approach with a limited evaluation.”**
> >
> > **A:** As the reviewer correctly points out, adapting DON and NDF to the PartNet evaluation is not straightforward, because both DON and NDF assume that the anchoring objects are stationary, whereas the anchor poses are varied in our PartNet-Mobility Placement task evaluation. We clarify this incompatibility in the text in Section 5.1 in the paper after listing the baselines. Regarding the additional object evaluations from NDF, we have added results for the bottle and bowl tasks in Supplement Section 5.3, Table S3. For these objects, we outperform NDF for mug and bowl for both upright and arbitrary poses, whereas NDF performs better for bottles. However, as previously mentioned, we do not require that anchor locations remain fixed as required by previous approaches.
> >
> > **Q: “The related work section … fails to mention the baselines compared to in the mug hanging experiment section, the most similar related work, as well as related ideas such as keypoint methods or canonicalization (e.g. NOCS). From the text alone, the novelty of the proposed method relative to these baselines is not clearly conveyed.”**
> >
> > **A:** We apologize for the omission of the relevant related works. We have added discussion of these approaches in the related works section in the updated version of the paper (attached to the new top post). Additionally we have added a description of language conditioned policies, such as CLIPort and Socratic Models, to the related works.
> >
> > We greatly appreciate your feedback, and please let us know if these changes have addressed your concerns and if there are other questions you may have!

---

### Author Response · Authors · 2022-08-28
**Revised paper + supplement v3**

**Comment:**

Revised paper and supplement version3. Updated parts are colored in magenta.

**Zip File:**

/attachment/d151f2dbee3dac6e8d6d84d99715b4da457a7c2b.zip

---

### Meta-Review · Area_Chair_MwAd · 2022-08-31

**Recommendation:** Accept (Poster)
**Confidence:** 3

**Metareview:**

This paper defines the cross-pose, the relative pose between two objects in an object manipulation task. End-to-end systems often fail to generalize to new objects. They create a vision-based system that learns to estimate cross-pose for a given task. Using cross-pose, a motion-based planner can be used to enable the robot to complete the task.

Strengths:

* Important problem that can enable generalization across multiple tasks with much less training data.
* Clear description and analysis of cross-pose.

Weaknesses:

* Cross-pose does not generalize when there is more than one possible start/end for the task.
* Missing some related work, and not clearly differentiated from some related work.